# Model Preserving Compression for Neural Networks

**Jerry Chee**[*]
Department of Computer Science
Cornell University
jerrychee@cs.cornell.edu

**Megan Flynn (née Renz)**[*]
Department of Physics
Cornell University
mr2268@cornell.edu

**Anil Damle**
Department of Computer Science
Cornell University
damle@cornell.edu

**Christopher De Sa**
Department of Computer Science
Cornell University
cdesa@cs.cornell.edu

## Abstract

After training complex deep learning models, a common task is to compress the model to reduce compute and storage demands. When compressing, it is desirable to preserve the original model's per-example decisions (e.g., to go beyond top-1 accuracy or preserve robustness), maintain the network's structure, automatically determine per-layer compression levels, and eliminate the need for fine tuning. No existing compression methods simultaneously satisfy these criteria—we introduce a principled approach that does by leveraging interpolative decompositions. Our approach simultaneously selects and eliminates channels (analogously, neurons), then constructs an interpolation matrix that propagates a correction into the next layer, preserving the network's structure. Consequently, our method achieves good performance even without fine tuning and admits theoretical analysis. Our theoretical generalization bound for a one layer network lends itself naturally to a heuristic that allows our method to automatically choose per-layer sizes for deep networks. We demonstrate the efficacy of our approach with strong empirical performance on a variety of tasks, models, and datasets—from simple one-hidden-layer networks to deep networks on ImageNet.

## 1 Introduction

There has recently been a significant theoretical and empirical effort to understand the necessity of over-parameterization for deep learning models and to develop algorithmic techniques that can effectively compress (or prune) them while retaining performance. This work is largely driven by results that show over-parameterization may aid in training via stochastic algorithms, but that the solutions found are often highly redundant [21, 22, 42, 54] and therefore can be compressed. Typically the focus has been on reducing parameter counts or FLOPs while maintaining accuracy—a task that often necessitates significant fine tuning of compressed models. However, we argue that further criteria are warranted to enable the adoption of compression methods amongst practitioners: (1) preserving the model's per-example decisions ensures the preservation of more fine-grain properties, e.g. fairness [4, 14, 17, 24, 61] or adversarial robustness [9, 10, 16, 18, 68]; (2) compressing networks while retaining their computational structure ensures compatibility with the rest of the machine learning pipeline; and (3) automatically determining per-layer sizing and minimizing the need for fine tuning alleviates key challenges when using compression methods in practice.

---

[*]equal contribution

36th Conference on Neural Information Processing Systems (NeurIPS 2022).

We develop a novel compression method that satisfies these three criteria. In particular, it uses unlabeled data to remove redundant neurons/channels while simultaneously updating the remaining network parameters to explicitly correct for the removed neurons/channels. The core technical development in our pruning method is the novel use of an interpolative decomposition (see Section 3) to approximate the post-activation output of layers. The interpolative decomposition (ID) explicitly represents the pruned neurons/channels as a linear combination of those kept—allowing us to preserve the model with little to no fine tuning. Moreover the specific structure of the ID allows us to sub-select channels backward through the activation function, and propagate information about the pruned channels forward, fusing it into the adjacent layer and preserving the structure of the network (albeit with smaller layer sizes). Collectively, these properties allow our ID based compression approach to produce a structurally similar compressed network that well approximates (to any desired accuracy) the original network without any additional training—a feature we show theoretically for simple fully connected networks in Section 4. We then extend the results to different layer types and more complex architectures in Section 5, and show how our theoretical results allow us to automatically determine per-layer sizes for a compressed model based on the trade-off between FLOPs and the estimated error induced by pruning.

We ensure that our method truly compresses a network and preserves the decision boundaries, rather than effectively re-training a smaller network through extensive fine tuning through a new metric we develop in Section 6. Consider pruning a pretrained VGG16 model on CIFAR-10 with $93.6\%$ accuracy to $50\%$ of its FLOPs. A variety of pruning methods achieve a reasonable accuracy of $93.2$-$93.6\%$, including structured magnitude pruning, random channel pruning, and our ID-based method. However, methods which rely extensively on fine-tuning (magnitude and random pruning) do not preserve the decision boundaries of the original network. The predictions of these pruned models agree with the original model only ($93.2\%, 93.2\%$ respectively) of the time on the test set. This level of agreement is barely better than a completely independently trained[2] (fullsize) VGG16 model ($93.0\%$). In contrast, our method produces a compressed model whose predictions are highly correlated with the original model, agreeing $97.3\%$ of the time.

To complement our algorithmic developments and theoretical contributions, in Section 7 we demonstrate the efficacy of our method on Atom3D [72], CIFAR-10 [43], and ImageNet [19]. The experiments are built to highlight how our ID-based compression method satisfies the desired criteria while competing methods do not. To accomplish this, we first introduce a metric of model similarity based on prediction correlation—this metric allows us to systematically determine how well compressed models are preserving the original model. To highlight the practical efficacy of our method (particularly its ability to automatically chose the size of the compressed network) we show our method successfully extends to Atom3D [72], simulating a real-world scenario with little prior compression work and baseline models with a different architecture (based on 3D convolutions). More generally, on all datasets we show our method has superior accuracy before fine tuning and is competitive with other state-of-the-art pruning methods after additional training is performed all while producing compressed models which better correlate with the original. Moreover, because our method only reduces layer sizes but otherwise preserves network structure, it can be easily composed with other compression methods. This allows us to substantially compress models for CIFAR-10 and ImageNet without the use of any global fine tuning.

In summary, the key contributions of this paper[3] are:

**A model preserving compression method** We introduce an ID-based algorithm for compressing neural networks that preserves the model's decisions and maintains the network's layer structure without the need for fine tuning.

**Theoretical guarantees** We provide theoretical guarantees for the output of our pruned model.

**Practical efficacy** We demonstrate the efficacy of our method across the ATOM3D, CIFAR-10 and ImageNet data sets and using a variety of models. Our method requires no labeled data, automatically determines per-layer sizes, and often requires little to no fine tuning.

---

[2]Consider a model $A$. Another model $B$ of the same architectures is *independently trained* if during training no information from $A$ is passed to $B$. Essentially this means retraining the same architecture from scratch twice to obtain models $A$ and $B$.

[3]Our code is available at https://github.com/jerry-chee/ModelPreserveCompressionNN

## 2 Related Work

There are a number of different design choices to be made in the compression and pruning process. Classical pruning involves eliminating either channels (analogously neurons) or individual weights, sometimes in a structured way. Magnitude pruning on both weights and neurons is still considered an effective approach [6, 21, 23, 25, 53, 46]. When pruning, the method may incorporate a correction to future layers, though it often does not [31, 54]. Some methods that correct the network chose to do local fine tuning [54, 32, 79, 66, 52], whereas others do not [48, 33].

Of particular note, matrix approximation methods [20, 40, 50, 65, 44] often satisfy the first criteria we desire for compression methods, but typically not the second as they add additional layers. These methods sometimes incorporate local fine tuning after compression [40, 42, 50, 65, 77] and sometimes do not [20, 44]. In contrast, structured pruning methods [31, 33, 48, 52-54] can satisfy the second criteria we outlined, but typically not the first as they do a poor job of preserving the model's decisions and often require excessive amounts of fine tuning.

Pruning with coresets [64] is the closest in spirit to our own work and provides a way to select a subset of neurons in the current layer that can approximate those in the next layer as well as new weight connections. Of note, Mussay et al. [64] provide a sample complexity result, and demonstrate their method on fully connected (but not convolution) layers. The HRank method [51] is also close in spirit to our own, and works by selectively pruning channels that produce low-rank feature maps. However, the method does not propagate updates into the next layer and instead relies on excessive amounts of fine tuning (30 epochs for each layer pruned) to fix the network's accuracy.

Recently the literature has started to consider criteria beyond topline accuracy metrics, and Liebenwein et al. [49] use measures of functional approximation to conclude that pruned networks well approximate the original models. Marx et al. [60] characterize when linear models can achieve similar accuracy but with competing predictions.

## 3 Interpolative decompositions

Our pruning strategy relies on a structured low-rank approximation known as an interpolative decomposition (ID). Classically, the Singular Value Decomposition (SVD) (see, e.g., [26]) provides an optimal low-rank approximation. However, because we consider matrices that include the non-linear activation function the SVD cannot be directly used to either subselect neurons or generate new ones (since it is unclear how to propagate singular vectors "backwards" through the non-linearity). In contrast, an ID constructs a structured low-rank approximation of a matrix $A$ where the basis used for the approximation is constrained to be a subset of the columns of $A$. For a matrix $A \in \mathbb{R}^{n \times m}$ we let $A_{\mathcal{J},\mathcal{I}}$ denote a sub-selection of the matrix $A$ using sets $\mathcal{J} \subset [n]$ to denote the selected rows and $\mathcal{I} \subset [m]$ to denote the selected columns; : denotes a selection of all rows or columns.

**Definition 3.1** (Interpolative Decomposition). Let $A \in \mathbb{R}^{n \times m}$ and $\epsilon \geq 0$. An $\epsilon$-accurate *interpolative decomposition* $A \approx A_{:,\mathcal{I}}T$ is a subset of columns of A, denoted with the index subset $\mathcal{I} \subset [m]$, and an associated interpolation matrix $T$ such that $\|A - A_{:,\mathcal{I}}T\|_2 \leq \epsilon \|A\|_2$.

*Remarks* 3.2. When computing an ID, we would like to find the smallest possible $k \equiv |\mathcal{I}|$ such that the accuracy requirement is satisfied. Moreover, we would like $T$ to have entries of reasonable magnitude and approximation error not much larger than the best possible for a given $k$ (i.e., $\|A - A_{:,\mathcal{I}}T\|_2 = t\sigma_{k+1}(A)$ for some small $t \geq 1$). While necessarily sub-optimal, the advantage is that we explicitly use a subset of the columns of $A$ to build the approximation.

IDs are well studied [13, 59], widely used in the domain of rank-structured matrices [35, 36, 34, 58, 57, 62], and are closely related to CUR decompositions [56, 74] and subset selection problems [7, 15, 73]. While these decompositions always exist, finding them optimally is a difficult task and in this work we appeal to what are known as (strong) rank-revealing QR factorizations [8, 11, 12, 28, 37].

**Definition 3.3** (Rank-revealing QR factorization). Let $A \in \mathbb{R}^{n \times m}$, $\ell = \min(n, m)$, and take any $k \leq \ell$. A *rank-revealing QR factorization* of A computes a permutation matrix $\Pi \in \mathbb{R}^{m \times m}$, an upper-trapezoidal matrix $R \in \mathbb{R}^{\ell \times m}$ (i.e. $R_{i,j} = 0$ if $i > j$), and a matrix $Q \in \mathbb{R}^{n \times \ell}$ with orthonormal columns (i.e., $Q^\top Q = I$) such that $A\Pi = QR$ and $Q$ and $R$ satisfy certain properties. Splitting $\Pi, Q$, and $R$ into $\Pi_1 \in \mathbb{R}^{m \times k}$, $\Pi_2 \in \mathbb{R}^{m \times (m-k)}$, $Q_1 \in \mathbb{R}^{n \times k}$, $Q_2 \in \mathbb{R}^{n \times (\ell-k)}$, $R_{11} \in \mathbb{R}^{k \times k}$,

$R_{12} \in \mathbb{R}^{k \times (m-k)}$, and $R_{22} \in \mathbb{R}^{(\ell-k) \times (m-k)}$ we can write

$$A \begin{bmatrix} \Pi_1 & \Pi_2 \end{bmatrix} = \begin{bmatrix} Q_1 & Q_2 \end{bmatrix} \begin{bmatrix} R_{11} & R_{12} \\ & R_{22} \end{bmatrix}. \tag{1}$$

*Remarks* 3.4. What makes (1) a rank-revealing QR factorization is that the permutation $\Pi$ is computed to ensure that $R_{11}$ is as well-conditioned as possible and $R_{22}$ is as small as possible. While more formal statements of these conditions exist, we omit them here as they do not factor into our work.

Critically, any rank-revealing QR factorization yields a natural rank-$k$ approximation of $A$ with error

$$\| A - Q_1 \begin{bmatrix} R_{11} & R_{22} \end{bmatrix} \Pi^\top \|_2 = \| R_{22} \|_2.$$

While finding the optimal rank-revealing QR factorization (*i.e.,* minimizing the error for a given $k$) is closely related to a provably hard problem [15], we find the original algorithm of Businger and Golub [8] works well in practice. This routine is available in LAPACK [2, 67], can be easily incorporated into existing code, and has computational complexity $\mathcal{O}(nmk)$ when run for $k$ steps.

**Computing interpolative decompositions**  Given a rank-revealing QR factorization, we can immediately construct an ID (a formal algorithmic statement is given in the appendix). Let $\mathcal{I} \subset [m]$ be such that $A_{:,\mathcal{I}} = A\Pi_1$ and define the interpolation matrix

$$T = \begin{bmatrix} I_k & R_{11}^{-1} R_{12} \end{bmatrix} \Pi^\top.$$

With the choice $A_{:,\mathcal{I}} = Q_1 R_{11}$ it follows that the error of the ID as defined by $\mathcal{I}$ and $T$ is $\| A - A_{:,\mathcal{I}} T \|_2 = \| R_{22} \|_2$. Picking $k$ such that $\| R_{22} \|_2 \leq \epsilon \| A \|_2$ yields the desired relative error. Notably, since $\kappa(A_{:,\mathcal{I}}) = \kappa(R_{11})$ and $T = \begin{bmatrix} I_k & R_{11}^{-1} R_{12} \end{bmatrix} \Pi^\top$ the desired criteria for an ID map back to those of a rank-revealing QR factorization—if $R_{11}$ is well conditioned then the basis we use for approximation is as well and entries of $T$ are not too large. If $\sigma_{\max}(R_{22})$ is not much larger than $\sigma_{k+1}(A)$ we get near optimal approximation accuracy.

**Accuracy of the matrix approximation**  A key feature of using a column-pivoted QR factorization to compute an ID is that it allows us to dynamically determine the approximation rank $k$ as a function of $\epsilon$. This can be accomplished by monitoring $\| R_{22} \|_2$ at each step of the column-pivoted QR algorithm. However, repeatedly computing $\| R_{22} \|_2$ is expensive and often unnecessary in practice. When using the algorithm by Businger and Golub [8] the magnitude of the diagonal entries of $R$ are non-increasing and it is common to use $|r_{k+1,k+1}/r_{1,1}|$ as a proxy for $\| R_{22} \|_2 / \| A \|_2$. While formal bounds indicate the approximation may be loose in the worst case, it is effective in practice (see appendix Figure 9) and once a candidate $k$ has been identified $\| R_{22} \|_2$ can be computed if desired to certify the result—if the accuracy is unacceptable $k$ can be increased until it is. In some settings it may be preferable to fix $k$ and simply accept whatever accuracy is achieved.

## 4  Pruning with interpolative decompositions

The core of our approach is a novel use of IDs to prune neural networks. Here we illustrate the scheme for a single fully connected layer and we extend the scheme to more complex layers (e.g., convolution layers) and deeper networks in Section 5. Consider a simple two layer (one hidden layer) fully connected neural network $h_{FC} : \mathbb{R}^d \to \mathbb{R}^c$ of width $m$ defined as

$$h_{FC}(x; W, U) = U^\top g(W^\top x)$$

with hidden layer $W \in \mathbb{R}^{d \times m}$, output layer $U \in \mathbb{R}^{m \times c}$, and activation function $g$. We omit bias terms: they may be readily incorporated by adding a row to $W$ and suitably augmenting the data.

To prune the model we will use an *unlabeled* pruning data set $\{x_i\}_{i=1}^n$ with $x_i \in \mathbb{R}^d$. Let $X \in \mathbb{R}^{d \times n}$ be the matrix such that $X_{:,i} = x_i$. Preserving the action of the two layer network to accuracy $\epsilon > 0$ on the data with fewer neurons is synonymous with finding an $\epsilon$ accurate approximation $h_{FC}(x; W, U) \approx h_{FC}(x; \widehat{W}, \widehat{U})$ where $\widehat{W}$ has fewer columns than $W$. We can do this by computing an ID of the activation output of the first layer.

Concretely, let $Z \in \mathbb{R}^{m \times n}$ be the first-layer output, i.e., $Z = g(W^\top X)$, and let $Z^\top \approx (Z^\top)_{:,\mathcal{I}} T$ be a rank-$k$ ID of $Z^\top$ with $|\mathcal{I}| = k$ and interpolation matrix $T \in \mathbb{R}^{k \times m}$ that achieves accuracy $\epsilon$ as in

Definition 3.1 (note that this means $k$ is a function of $\epsilon$). Because the activation function $g$ commutes with the sub-selection operator, if $Z \approx T^\top Z_{\mathcal{I},:}$ then

$$g(W^\top X) \approx T^\top \left(g(W^\top X)\right)_{\mathcal{I},:} = T^\top g\left(W_{:,\mathcal{I}}^\top X\right).$$

Multiplying both sides by $U^\top$ now gives an approximation of the original network by a pruned one,

$$h_{FC}(x; W, U) = U^\top g(W^\top X) \approx h_{FC}(x; W_{:,\mathcal{I}}, TU) = U^\top T^\top g\left(W_{:,\mathcal{I}}^\top X\right). \qquad (2)$$

That is, the ID has pruned the network of width $m$ into a dense sub-network of width $k$ with $\widehat{W} \equiv W_{:,\mathcal{I}} \in \mathbb{R}^{d \times k}$ and $\widehat{U} \equiv TU \in \mathbb{R}^{k \times c}$. Importantly, the SVD of $Z^\top$ cannot be used for this task since it is not clear how to map the dominant left singular vectors back through the activation function $g$ to either a subset of the existing neurons or a small set of new neurons. This makes use of the ID essential and we provide additional intuition for this scheme in the appendix, specifically in Figure 7.

## 4.1 A generalization bound for the pruned network

A key feature of our ID based pruning method is that it can be used to dynamically select the width of the pruned network to maintain a desired accuracy when compared with the full model. This allows us to provide generalization guarantees for the compressed network in terms of generalization of the full model and the accuracy of the ID. We state the results for a single hidden fully connected layer with scalar output (i.e., $c = 1$) and squared loss. They can be extended to more complex networks (at the expense of more complicated dependence on the accuracy each layer is pruned to), more general Lipschitz continuous loss functions, and vector valued output. We defer all proofs to the supplementary material.

Assume $(x, y) \sim \mathcal{D}$ where $x \in \mathbb{R}^d$, $y \in \mathbb{R}$, and the distribution $\mathcal{D}$ is supported on a compact domain $\Omega_x \times \Omega_y$. We let $\mathcal{R}_0 = \mathbb{E}_{(x,y)\sim\mathcal{D}}(\|(u^\top g(W^\top x) - y)\|^2)$ denote the true risk of the trained full model and $\mathcal{R}_p = \mathbb{E}_{(x,y)\sim\mathcal{D}}(\|(\widehat{u}^\top g(\widehat{W}^\top x) - y)\|^2)$ be the risk of the pruned model. (Since $c = 1$ we let $u \in \mathbb{R}^m$ denote the last layer.) We also define the empirical risk $\widehat{\mathcal{R}}_{ID}$ of approximating the full model with our pruned model as

$$\widehat{\mathcal{R}}_{ID} = \frac{1}{n} \sum_{i=1}^{n} \left|u^\top g(W^\top x_i) - \widehat{u}^\top g(\widehat{W}^\top x_i)\right|^2,$$

where $\{x_i\}_{i=1}^n$ are $n$ i.i.d. samples from $\mathcal{D}$ (note that we do not need labels for these samples). Using this notation, Theorem 4.1 controls the generalization error of the pruned model.

**Theorem 4.1** (Single hidden layer FC). *Consider a model $h_{FC} = u^\top g(W^\top x)$ with $m$ hidden neurons and a pruned model $\widehat{h}_{FC} = \widehat{u}^\top g(\widehat{W}^\top x)$ constructed using an $\epsilon$ accurate ID with $n$ data points drawn i.i.d from $\mathcal{D}$. The risk of the pruned model $\mathcal{R}_p$ on a data set $(x, y) \sim D$ assuming $\mathcal{D}$ is compactly supported on $\Omega_x \times \Omega$ is bounded by*

$$\mathcal{R}_p \leq \mathcal{R}_{ID} + \mathcal{R}_0 + 2\sqrt{\mathcal{R}_{ID}\mathcal{R}_0}, \qquad (3)$$

*where $\mathcal{R}_{ID}$ is the risk associated with approximating the full model by a pruned one and with probability $1 - \delta$ satisfies*

$$\mathcal{R}_{ID} \leq \epsilon^2 M + M(1 + \|T\|_2)^2 n^{-\frac{1}{2}} \left(\sqrt{2\zeta dm \log(dm) \log \frac{en}{\zeta dm \log(dm)}} + \sqrt{\frac{\log(1/\delta)}{2}}\right).$$

*Here, $M = \sup_{x \in \Omega_x} \|u\|_2^2 \|g(W^T x)\|_2^2$ and $\zeta$ is a universal constant that depends on $g$.*

Theorem 4.1 is developed by considering the ID as a learning algorithm applied to the output of the full model using unlabeled pruning data. This allows us to control the risk of the pruned model in terms of the risk of the original model and the additional risk introduced by the ID. Importantly, here we can control the additional risk in terms of the empirical risk of the ID and an additive term that decays as additional pruning data is used. Lemma 4.2 codifies this decomposition.

**Lemma 4.2.** *Under the assumptions of Theorem 4.1, for any $\delta \in (0, 1)$, $\mathcal{R}_{ID}$ satisfies*

$$\mathcal{R}_{ID} \leq \widehat{\mathcal{R}}_{ID} + M(1 + \|T\|_2)^2 n^{-\frac{1}{2}} \left(\sqrt{2p \log(en/p)} + 2^{-\frac{1}{2}}\sqrt{\log(1/\delta)}\right)$$

*with probability $1 - \delta$, where $M = \sup_{x \in \Omega_x} \|u\|^2 \|g(W^T x)\|^2$ and $p = \zeta dm \log(dm)$ for some universal constant $\zeta$ that depends only on the activation function.*

*Remarks* 4.3. We believe that the second part of the bound in Lemma 4.2 is likely loose since it relies on a pseudo-dimension bound for fully connected neural networks. However, when pruning with an ID we only consider subsets of existent neurons and it is plausible that in this setting the upper bound for the pseudo-dimension could be improved.

Crucially, an immediate consequence of using an ID for pruning is that we can explicitly control $\mathcal{R}_{ID}$ in terms of the accuracy parameter. This relation between the ID accuracy and empirical risk is given in Lemma 4.4 and is what allows us to express the risk of the pruned network in Theorem 4.1.

**Lemma 4.4.** *Following the notation of Theorem 4.1, an ID pruning to accuracy $\epsilon$ yields a compressed network that satisfies $\widehat{\mathcal{R}}_{ID} \leq \epsilon^2 \|u\|_2^2 \|g(W^T X)\|_2^2 / n$, where $X \in \mathbb{R}^{d \times n}$ is a matrix whose columns are the pruning data.*

## 5 Convolutional and deep networks

### 5.1 Convolution layers

To prune convolution layers with the ID at the channel level we reshape the output tensor into a matrix where each column represents a single output channel. After this transformation, the key idea is the same as in Section 4. Consider a simple two layer convolution neural network defined as $h_{\text{conv}}(x; W, U) = \text{conv}(U, g(\text{conv}(W, x))))$, with the convolution-layer operator conv, weight tensors $W$ and $U$, and activation function $g$. The kernel dimension, size, stride, padding, and dilation do not change the form of the ID at the output channel level. Let $Z = g(\text{conv}(W, X))$ be the activation output of the first layer with unlabeled pruning data $X$, and define reshape as the operator which reshapes a tensor into a matrix with the output channels as columns, i.e., $\text{reshape}(Z) \in \mathbb{R}^{n_i \times m_c}$ where $m_c$ is the number of channels and $n_i$ is the product of all other dimensions (e.g. in the case of a 2d convolution, $n_i$ would be width $\times$ height $\times$ number of examples).

We now compute[4] a rank-$k$ ID $\text{reshape}(Z) \approx \text{reshape}(Z)_{:, \mathcal{I}} T$. The activation function $g$ and reshape operator both commute with the sub-selection operator, so

$$\text{reshape}(g(\text{conv}(W, X))) \approx \text{reshape}(g(\text{conv}(W, X)))_{:, \mathcal{I}} T$$
$$= \text{reshape}(g(\text{conv}(W_{\mathcal{I}, \ldots}, X)))T,$$

where $W_{\mathcal{I}, \ldots}$ denotes an indexing sub-selection of $W$ along its output-channel dimension. Next, we need to propagate this $T$ into the next layer, which we can do with a "matrix multiply" by the next-layer's weights along its input channel dimension: we call this operation matmul.[5] With this, a little algebraic manipulation of our approximate equality above gives

$$\text{conv}(U, g(\text{conv}(W, X)))) \approx \text{conv}(\text{matmul}(T, U), g(\text{conv}(W_{\mathcal{I}, \ldots}, X)))), \tag{4}$$

and so if we set $\widehat{U} = \text{matmul}(T, U)$ and $\widehat{W} = W_{\mathcal{I}, \ldots}$, we can preserve the action of the two layer network with fewer channels as $h_{\text{conv}}(x; W, U) \approx h_{\text{conv}}(x; \widehat{W}, \widehat{U})$. This gives us a recipe for pruning convolution layers analogous to (2). This recipe can be directly applied to a composition of a convolution layer followed by a pooling layer [27] (or any other layer that acts independently by channel) by treating the conv layer/ pooling layer pair as a single convolution layer with a "fancy" activation function $g$: we just run the ID on the output post-pooling, and use that to sub-select the convolution layer's weights. Flatten layers, for connecting to FC layers, are equally straightforward.

### 5.2 Deep networks

The ID pruning primitives for fully connected and convolution layers can now be composed together to prune deep networks. Algorithm 1 specifies how we chain together the fully connected and convolution primitives to prune feedforward networks, for simplicity we assume for the moment we know the desired layer sizes. A multi-layer neural network is pruned from the beginning to the end, where the ID is used to approximate the outputs of the original network. The ID pruning primitives

---

[4]When $n_i$ is large we can appeal to randomized ID algorithms [59], or the TSQR [3].

[5]If $T \in \mathbb{R}^{m \times n}$ and $U$ is a weight-tensor with $n$ input channels, then to compute $\text{matmul}(T, U)$ we: (1) reshape $U$ to be an $n \times p$ matrix for some $p$, (2) multiply the reshaped matrix by $T$, producing a $m \times p$ matrix, and (3) reshape the result back to a tensor with $m$ input channels and all other dimensions the same as $U$.

---

**Algorithm 1** Pruning a multilayer network with interpolative decompositions

---

**Input:** Neural net $h(x; W^{(1)}, \ldots, W^{(L)})$, layers to not prune $S \subset [L]$, pruning set $X$, pruning fraction $\alpha$
**Output:** Pruned network $h(x; \widehat{W}^{(1)}, \ldots, \widehat{W}^{(L)})$

1:  $T^{(0)} \leftarrow I$
2:  **for** $l \in \{1 \ldots L\}$ **do**
3:      $Z \leftarrow h_{1:l}(X; W^{(1)}, \ldots, W^{(l)})$ {layer l activations}
4:      **if** layer $l$ is a FC layer **then**
5:          $(\mathcal{I}, T^{(l)}) \leftarrow \text{ID}(Z^T; \alpha)$ **if** $l \notin S$ **else** $(:, I)$
6:          $\widehat{W}^{(l)} \leftarrow T^{(l-1)} W^{(l)}_{:,\mathcal{I}}$ {sub-select neurons, multiply T of prev layer's ID}
7:      **else if** layer l is a Conv layer (or Conv+Pool) **then**
8:          $(\mathcal{I}, T^{(l)}) \leftarrow \text{ID}(\text{reshape}(Z); \alpha)$ **if** $l \notin S$ **else** $(:, I)$
9:          $\widehat{W}^{(l)} \leftarrow \text{matmul}(T^{(l-1)}, W^{(l)}_{\mathcal{I},\ldots})$ {select channels; multiply T}
10:     **else if** layer l is a Flatten layer **then**
11:         $T^{(l)} \leftarrow T^{(l-1)} \otimes I$   {expand T to have the expected size}
12:     **end if**
13: **end for**

---

sub-select neurons (or channels) in the current layer and propagate the corrective interpolation matrix to the next layer. There are many ways one could prune a multi-layer network with these ID pruning primitives; we selected the approach in Algorithm 1 through empirical observations (though we do not assert that it is optimal). Note that as a pre-processing step before running Algorithm 1, batch normalization layers [41] should be absorbed into their preceding fully-connected or convolution layers, and dropout [70] layers should be removed.

**Iterative Pruning**   While Algorithm 1 is illustrative, in practice we would often like to be able to either specify a desired accuracy or choose layer sizes optimally for a desired compression ratio. Our approach allows us to accomplish this by iteratively selecting layers to compress. We introduce a score function for layers that is the ratio of the estimated relative error $|r_{k+1,k+1}/r_{1,1}|$ introduced by compressing a layer to the number of flops $f_l$ that would be cut if we pruned a layer $l$ to size $k$. We call this score $s_l(k) = |r_{k+1,k+1}/r_{1,1}|/f_l$ and it is heuristic for the compressability of each layer—lower scores imply a layer is easier to compress. However, different layers of the network are connected, and compressing a layer early in the network can effect how well later layers can be compressed. Therefore, we prune the network iteratively, measuring the score for each layer at a given pruning percentage (or step size), choosing the layer with the lowest score, pruning it, and then re-calculating the scores for the later layers. We repeat the process until the network reaches a desired compression, or until the network performance degrades unacceptably. We refer to this method as Iterative ID, and refer to cutting a constant fraction of all neurons in each layer as Constant Fraction ID. For full details see Appendix F and Algorithm 3.

## 6   Evaluating compression beyond accuracy

An important benefit of our approach is that we are actually able to preserve the original model's predictions better than other methods. Traditional pruning methods typically do a poor job at preserving the original predictions, due to their heavy reliance on fine tuning that effectively retrains the model. Here we explain why we might care about preserving per-example predictions beyond top-line accuracy. We argue that in many situations compression methods must well-approximate the original model, and that accuracy is a poor metric for this use case.

Consider a pretrained model $M$, a resulting "compressed" model $M_C$, and an evaluation set $(X, Y)$. Accuracy measures the similarity between the true labels $Y$ and the predicted labels from the compressed model $M_C(X)$. This metric does not directly compare the original model $M$ and compressed model $M_C$. As we have seen in Section 1, a compressed model can recover the accuracy of the original model, but still differ widely on the predictions. Instead our model correlation measures the similarity between the original model's predictions $M(X)$ and the compressed model's predictions $M_C(X)$. One can think of this metric as an "accuracy to the original predictions", instead of an accuracy to the true labels.

We propose model correlation as the percent of test example predictions two models agree on—details of this metric are discussed in Appendix D. Model correlation is a general metric to measure the similarity between the learned functions of two models. In other contexts people have explored the related concept of model fidelity, where our correlation metric matches the top-1 agreement of Stanton et al. [71]. Our claim is that by better preserving a model's per-example decisions, we can better preserve special properties of the model. In the following section we provide experimental evidence for this claim. Models are now often trained to have properties that go beyond test accuracy—for example robustness to adversarial attacks, sub-class classification accuracy, fairness, etc., and this measure of model correlation is likely to correlate with many of these criteria. Note that we do not believe preserving per-example decisions "boosts" any of these properties, we are simply preserving properties of the baseline model.

## 7 Experiments

The Appendix provides a list of all methods we compare with (including references) and a lookup table to find the relevant experiments for each method (i.e., which figures and tables they appear in).

**More careful evaluation of pruning methods** We propose a range of evaluations that goes beyond simple post-fine tuning test accuracy on standard vision benchmarks. First is our proposed model correlation metric, discussed in detail in Section 6. While certain benchmark problems are well studied and the literature has characterized layer sizes for more efficient networks, that cannot be relied upon for realistic problems practitioners want to solve. Therefore, we demonstrate the usability of our method by applying it to a non-standard problem and automatically discover a significantly more efficient allocation of flops per layer. We also consider extensive pre-fine tuning evaluations. In many cases, we may produce results before fine tuning that are sufficiently strong to remove the need for it. Moreover, fine tuning often washes out differences between different methods and reduces model correlation (particularly if the learning rate is too high for too many epochs). We show that by preserving model correlation our method is able to preserve the sub-class accuracy of a class which has been removed from the pruning and fine tuning set. Lastly, we show how maintaining network structure allows us to compose our methods with other matrix-decomposition based techniques and provide post-fine tuning results. One hidden layer results can be found in the Appendix.

**Broad applicability** We demonstrate the versatility of our method on atypical architectures and beyond vision tasks using the ATOM3D Ligand Binding Affinity (LBA) [72] benchmark. This simulates a more typical use case for pruning than benchmark vision models. The LBA task predicts the binding strength between proteins and small molecules which is useful for drug discovery. We train and prune the 3DConv network from [72], which uses 3-D convolutional layers, and achieves a RMSE of 1.42. We use the same method to achieve a baseline RMSE of 1.43, and prune the network using our Iterative ID. The baseline network uses 12.8G FLOPs per example. For comparison VGG-16 on Imagenet uses 30G FLOPs. Figure 1 shows that Iterative ID is able to prune 95% of the total FLOPS without any significant change in the accuracy of the network and does not require any fine tuning.

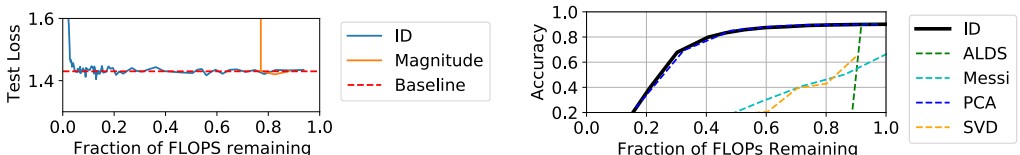

Figure 1: Pre-fine tuning pruning results on a 3D-Conv network for ATOM3D (left) and MobileNet V1 for CIFAR-10 (right). We see that ID either matches or outperforms other methods.

**Pre-fine tuning results** We analyze the performance of ID compression against other pruning methods using the CIFAR-10 [43] and Imagenet [19] data sets with VGG16 [69] and Mobilenet V1 [38]. The ID uses a held-out pruning set of 1000 data points on CIFAR-10 and 5000 data points on Imagenet. Full hyper-parameter details can be found in the Appendix and code. Figures 1 and 2 illustrate results before fine tuning for structured and unstructured methods. We see that on CIFAR-10

and ImageNet, Iterative ID matches or outperforms other methods which do not use any global fine tuning. We even dominate unstructured methods which are typically more parameter efficient. Many matrix decomposition based methods struggle to compress the depth-wise-separable convolution layers in the Mobilenet V1 network. Moreover, Iterative ID is able to choose better per-layer sizes than the default VGG16 configuration on CIFAR-10. In the appendix, we show that the choice of how much to prune per iteration is not particularly sensitive, and demonstrate that pruning up to 35% of the FLOPs results in no degradation to the pre-fine tuning accuracy. Results for MobileNet V1 on ImageNet can be found in the Appendix.

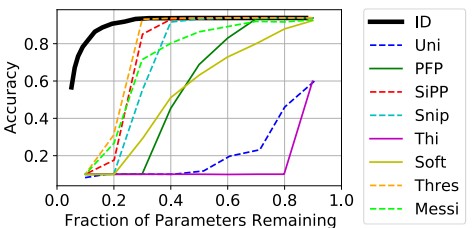 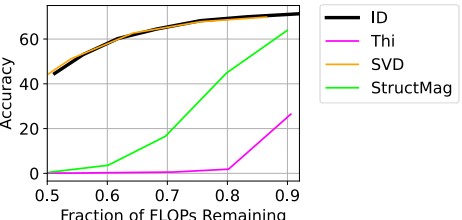

Figure 2: Pre-fine tuning pruning results on VGG16 for CIFAR-10 (left) and ImageNet (right). ID either matches or outperforms other methods, including unstructured methods (dotted lines) which are typically more parameter efficient. Solid lines are structured. Note the "SVD" method changes the structure of the network. Model correlation results can be found in the Appendix.

**Model correlation as a proxy for preserving fairness** We conduct an experiment to demonstrate the connection between model correlation and a simple fairness metric of per-class accuracy. We begin with a VGG16 model $M$ trained on all 10 classes of Cifar10. It performs reasonably well on all 10 classes, so we consider it "fair". We then remove a class from the pruning set to simulate an under-represented class (but leave it in the train and test sets). We then prune $M$ using our ID-based method, creating a compressed model $M_{id}$—we do not perform any fine tuning. Sub-class accuracy for each class can be seen in figure 14a as well as the correlation of $M_{id}$ with respect to $M$. Despite not having access to an entire class of data, we can prune the model to 50% FLOPs while only losing a few percent accuracy on the missing class and retaining reasonably high model correlation. As a comparison we prune the model $M$ with magnitude pruning to create a compressed model $M_m$. The magnitude pruned model loses almost all accuracy on the missing class when we prune to 50%, and its model correlation is initially only 10%. We fine tune with the abbreviated data set in an attempt to recover accuracy (figure 15a shows the correlation and sub-class accuracies). While $M_m$ recovers on the represented classes, it fails to recover any accuracy for the unrepresented class during fine tuning.

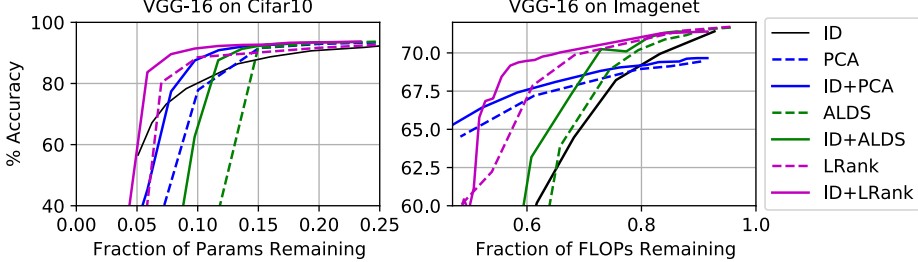

Figure 3: Composing ID with other compression methods for VGG-16 on CIFAR-10 (left) and ImageNet (right). For each ID+compression method, the original network was pruned using ID until the correlation degraded on the pruning set. A second method was then applied to the compressed network. This composition produced smaller and more accurate networks than either method alone.

**Model architecture preservation and composition of methods** Because our method preserves network structure and only reduces layer sizes it can be easily combined with other methods, see Figure 3. We first apply Iterative ID , pruning until the the model correlation begins to drop on the pruning set. Then we apply a secondary compression method. This composition is easy because we are simply replacing the original model with an ID-pruned model that has the same architecture.

The secondary algorithms we use insert extra layers with different computational structure into the network, so composing them with each other is non-trivial. In each example we see that pruning first with ID and then applying a secondary algorithm boosts the performance of the secondary algorithm.

**Post-global-fine tuning results** In Tables 1, and 2 we demonstrate the effectiveness of our method for VGG-16 on CIFAR-10 and ImageNet after moderate amounts of global fine tuning, both alone and when combined with other methods. Hyperparameter details can be found in the Appendix. Networks pruned with ID attain competitive accuracies while having a much higher prediction correlation with their original models. This evidence suggests that we have truly compressed the network rather than retraining a new one. Other methods often do not correlate much more than an independently trained model. Surprisingly, we find that in some settings combining ID with other matrix decomposition methods like LRank[40] can be competitive without any global fine tuning (Tables 1, 2).

We also find that ID pruning and then globally fine-tuning a MobileNet V1 architecture results in an accuracy of 89.4% on Cifar-10 and a correlation of 92.6% with the original model after 60 epochs at a 42% FLOPs reduction, as opposed to a model trained from scratch which achieves an accuracy of 88.9% and a correlation of 88.6% after 120 epochs.

| Method | FLOPs Reduction (%) | Accuracy | $\Delta$ | Correlation | Epochs |
|---|---|---|---|---|---|
| ID | 50 | 93.31 | 0.30 | 97.29 | 60 |
| ID+LRank | 50 | 93.43 | 0.31 | 95.9 | 0 |
| HRank | 54 | 93.43 | -0.50 | — | 480 |
| Polar | 54 | 93.92 | 0.04 | — | 200 |
| NS | 51 | 93.62 | -0.26 | — | 200 |
| Mag | 50 | 93.56 | -0.04 | 93.15 | 200 |
| Independent | 0 | 93.60 | — | 93.02 | 160 |

Table 1: Post-fine tuning VGG-16 results on CIFAR-10. ID results are averaged over 5 trials, with a standard deviation of $\pm 0.36$. "Independent" is an independently trained model. In the 50% FLOPs region, ID produces the most correlated model, while ID+LRank produces a model with reasonable accuracy without any global fine tuning. $\Delta \equiv$ original accuracy - pruned accuracy.

| Method | FLOPs Reduction (%) | Accuracy | $\Delta$ | Correlation | Epochs |
|---|---|---|---|---|---|
| ID+LRank | 43 | 68.84 | 2.67 | 85.66 | 0 |
| ID+LRank | 43 | 70.50 | 1.01 | 90.90 | 10 |
| ID+PCA | 54 | 69.18 | 2.33 | 87.13 | 200 |
| LRank([50]-impl.) | 43 | 70.30 | 1.21 | 90.38 | 10 |
| ThiNet | 46 | 66.59 | 0.89 | 70.15 | 30 |
| AMC | 48 | 70.49 | 0.65 | 76.03 | 120 |
| CP | 53 | 68.20 | 2.66 | 78.47 | 10 |

Table 2: Post-fine tuning VGG-16 results on ImageNet. We list methods which release both the original and compressed model, so that we can compute the correlation. ID composed methods produce competitive accuracies at a range of FLOPs reductions, and produce better correlated models, even without any fine tuning.

## 8 Limitations and future work

Our method comes with several limitations and possible extensions for future work. While preserving model correlation suggests that we are likely to preserve sub-class loss, our theory does not currently extend to that regime and requires that the unlabeled pruning set be from the same distribution as the test set. More broadly, we do not yet fully understand how trainable the models produced by ID are in different conditions and we cannot make claims about how compressable a given model will be. In future work we will explore modifying training process to improve prunability, which is a common approach [39, 76, 52, 78]. We will also explore ways to refine our iterative pruning approach to work with more complicated architectures. Of particular note, our method typically exposes a sizable "free FLOPs" regime and we explore how this can be leveraged more broadly.

### Acknowledgments and Disclosure of Funding
This work was partially funded by the National Science Foundation under awards DMS-1830274, DGE-1922551, DMS-2146079, and NSF CAREER Award 2046760.

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
