# OpenReview forum: "Model Preserving Compression for Neural Networks"
_NeurIPS.cc/2022/Conference — NeurIPS 2022 Accept_

### Official Review · Reviewer_12sE · 2022-07-10

**Rating:** 2
**Confidence:** 5
**Soundness:** 1 poor
**Presentation:** 2 fair
**Contribution:** 2 fair

**Summary:**

This paper proposes a pruning approach based on interpolative decomposition computed by rank revealing QR. The proposed method can automatically determine the layer size. The authors also provide theoretical analysis for the proposed methods.

**Questions:**

1. There is no explanation about ID+LRank. LRank decomposes layers into a low-rank format using an alternating Lagrangian algorithm, learning to compression. How does it combine with the proposed ID-based pruning approach?
2. What is the performance gap between the pruned model and the pre-trained model? All the results miss baselines.
3. How does this approach perform for SOTA models on ImageNet?

**Limitations:**

To make the proposed approach convincing, I highly recommend the authors to provide more general experimental results for fair comparison, e.g., ResNet-50, MobileNet and ViT on ImageNet. I also recommend the authors to not recycle to another top-tier conference without significant revision or improvement.

**Strengths And Weaknesses:**

Strengths:
1. Using interpolative decomposition is new to compression and it is interesting to maintain the pretrained model information with ID.
2. Pruning from a pre-trained model without retraining is practically useful.
3. The motivation of this paper is good and the theoretical guarantee is significant.

Weaknesses:
1. The experiments are too weak. VGG is an extremely dated model. Results and comparisons for ResNet-50 and MobileNet on ImageNet are required. If authors could provide results on vision transformers, it could be more convincing.
2. Even though the theoretical parts are promising, the practical results are weak. It is not sure whether the proposed post-training pruning method really works. Based on Table 2,  the performance is even inferior to AMC.
3. The metric 'correlation' used in this paper is not a general metric to evaluate the model performance. Better correlation shows nothing.
4. This paper lacks several SOTA references, e.g., [R1-R4].

[R1] Scop: Scientific control for reliable neural network pruning. NeurIPS 2020.

[R2] ChipNet: Budget-Aware Pruning with Heaviside Continuous Approximations. ICLR 2021.

[R3] Provable filter pruning for efficient neural networks, ICLR 2020.

[R4] Compressing Neural Networks: Towards Determining the Optimal Layer-wise Decomposition, NeurIPS 2021.

---

> ### Author Response · Authors · 2022-08-02
> **Response to reviewer 12sE**
>
> Thank you for the feedback; we will address the concerns point by point. Our main goal is to present a new idea in a clear setting. There are further extensions we are pursuing, some of which coincide with those suggested by the review. However, we believe that our novel method, theoretical analysis, and set of empirical evaluations provide a cohesive and complete piece of work on this method.
>
> Q: *Experiments for ResNet-50 and Mobilenet and Imagenet are required. Results on image transformers [would be] more convincing.* \
> A: Please see Figure 1(right) and Line 328 for MobileNet results in our main paper, and Figures 10 and 11 in the appendix for additional MobileNet experiments, including on ImageNet. Due to space limitations we had to move some MobileNet experiments to the appendix. \
> Our compression method is not specialized to ResNets and we chose architectures that allow us to present our ideas in a clear way. In future work we can look into these extensions (including vision transformers), but we believe that the slate of experiments chosen here (including on “non standard” models and problems to demonstrate breadth and ease of use) provide good support for our novel procedure. In addition, none of the suggested references R1-R4 include experiments on vision transformers in the main body of their papers.
>
>
> Q: *The performance is inferior to AMC (Table 2).* \
> A: AMC uses 10X the number of fine tuning epochs. It also is much worse at preserving the original pre-trained network.
>
> Q: *Better correlation shows nothing.* \
> A: We respectfully disagree.  Simple accuracy metrics are but a single property of large, complex deep learning models. While important in many settings, accuracy also does not fully capture the efficacy (or lack thereof) for certain applications—something that is becoming increasingly clear. Model correlation provides an additional metric for evaluating performance and can be important in cases where the baseline models have “features” (e.g., controlling the types of errors that are made) that the pruned model should preserve. Concisely, model correlation helps show if we are actually computing a compression of the baseline model rather than simply finding a smaller model that achieves similar accuracy.
>
> Q: *The paper lacks SOTA references [R1-R4].* \
> A: Respectfully, the statement that we did not compare to R3 and R4 is incorrect.  We have cited R3 as reference 49 and R4 as reference 51, and have direct comparisons to those two references in Figure 1 (right), Figure 2 (left) and Figure 3.
> We would like to highlight that we have included a table of citations (organized by pruning method) and corresponding figure/table numbers in the Appendix under Section E.  We hope this table can help clarify the breadth of our comparisons and aid in finding specific comparisons.
>
> R1 and R2 evaluate only on ResNets or MobileNetV2s and thus we have no way to directly compare them to our method (note we use the MobileNetV1 not V2). In addition, both use 300 or more epochs of fine tuning and in this work we focus on understanding how performant methods can be when little to no fine tuning is used. We can add R1 and R2 to our related work.
>
> Q: *There is no explanation about ID+LRank.* \
> A: Please see our explanation in the paragraph starting at Line 311. In brief, our method outputs a smaller model with the same architecture as the original model (i.e., no additional layers are added). Therefore, the output from our method can be passed to LRank (or any other method) as if it was the baseline model and no modifications to the follow-up method are required.  In rather terse pseudo code, this is analogous to writing:
>
> M = original model \
> M = ID_compress(M) \
> M = LRrank_compress(M)
>
>
> Q: *What is the performance gap between pruned and pre-trained models?* \
> A: For Tables 1 & 2, please see the “delta” column to the right of the “Accuracy” column for the difference in accuracy to the baseline. We will make sure to label this more clearly in the Table caption. For the Figures, the accuracy of the pre-trained models corresponds to the FLOPs or Parameters Remaining = 100%. We will make sure to add a horizontal line in the figures to make this more clear.
>
> Q: *How does this approach perform for SOTA models on ImageNet?* \
> A: Can you provide specific references? We compare to several methods on ImageNet in Figures 2, 3 and Table 2, as well as in section H in the Appendix. Many of our comparisons were published in the past 2 years.

---

> > ### Comment · Reviewer_12sE · 2022-08-05
> > **Most concerns are not addressed especially practical performance**
> >
> > I would like to thank all the authors for their response. I do appreciate you for providing elegant theoretical content. However, if you are not able to provide practical evidence that your method works well on large-scale datasets for sophisticated models, I won't consider the proposed approach as novel.
> >
> > Regarding experiments:
> >
> > Experiments for ResNet-50 and MobileNetV2 on ImageNet are the most important and common ones in compression, while VGG-16 is an ancient model which is extremely redundant and CIFAR-10 is a toy dataset. I have never seen any paper misses ResNet-50 since 2021 [R4-R13]. Even before 2021, I believe most papers provide this experiment [R14-R18]. I totally disagree that authors can get rid of ResNet by just saying your proposed method is not suitable for ResNet. Besides, All the existing models contain residual connections, which stem from ResNet. As you claim your approach works well for MobileNetV1, isn't it weird that it does not work for ResNet? If you really want to get rid of ResNet, please provide results for MobileNetV2 on ImageNet, which is the most common experiment in recent papers, and directly cite SOTA's reported number. If you want to compare with ALDS [R4], please run the same experiments (e.g., ResNet-18 and MobileNetV2) instead of testing MobileNetV1 that is NOT reported in ALDS at all.
> >
> > Let us look back at what authors provide. Based on Figure 2 (right), where the exact accuracy number is extremely NOT clear, for VGG-16, I have to try my best to guess that the accuracy is 60% when 40% FLOPs are removed. However, as reported in [R8], they remove 52.4% FLOPs or 93.9% parameters, achieving 68.81% accuracy, which is 9% higher than this paper. Additionally, [R13] reports even AMC can achieve 69.10% top-1 when removing 80% FLOPs. I don't know how this paper could claim the proposed method works well practically. Let us move to MobileNetV1 part. I can't believe the top-1 performance of ALDS is less than 20% while 90% FLOPs remain. I highly suspect that you run their code incorrectly. The fact is that you have indeed compared with ALDS, but the terrible results are NOT reported in the original paper.
> >
> > Regarding ID+LRank:
> >
> > LRank needs two stages of compression, which is quite different from others. At the first stage, it trains the uncompressed model with an augmented Lagrangian method, imposing low-rank properties on the pre-trained model. At the second stage, it decomposes the uncompressed model then retrains the compressed model. You claim your method does not need post training. I am still not sure how to achieve your ID+LRank. I don't think L311-L319 explains it. Besides, I don't know the purpose of showing the results of such combinations (ID+others). This is not consistent with the contribution claimed in Intro.
> >
> > Correlation:
> >
> > You could claim this metric is useful. However, for FAIR comparison, I prefer to care about the general metrics to evaluate the performance, which are already recognized by researchers in this community. When your method practically achieves similar performance of SOTAs, this 'correlation' could be an auxiliary trick to explain.
> >
> >
> > [R5] GDP: Stabilized Neural Network Pruning via Gates with Differentiable Polarization, ICCV 2021
> >
> > [R6] Network Pruning via Performance Maximization, CVPR 2021
> >
> > [R7] Manifold Regularized Dynamic Network Pruning, CVPR 2021
> >
> > [R8] Towards Compact CNNs via Collaborative Compression, CVPR 2021
> >
> > [R9] CHIP: CHannel independence-based pruning for compact neural networks, NeurIPS 2021
> >
> > [R10] Revisiting Random Channel Pruning for Neural Network Compression, CVPR 2022
> >
> > [R11] When To Prune? A Policy Towards Early Structural Pruning, CVPR 2022
> >
> > [R12] DepthShrinker: A New Compression Paradigm Towards Boosting Real-Hardware Efficiency of Compact Neural Networks, ICML 2022
> >
> > [R13] Topology-Aware Network Pruning using Multi-stage Graph Embedding and Reinforcement Learning, ICML 2022
> >
> > [R14] Group Sparsity: The Hinge Between Filter Pruning and Decomposition for Network Compression, CVPR 2020
> >
> > [R15] PENNI: Pruned Kernel Sharing for Efficient CNN Inference, ICML 2020
> >
> > [R16] HRank: Filter Pruning using High-Rank Feature Map, CVPR 2020
> >
> > [R17] DSA: More Efficient Budgeted Pruning via Differentiable Sparsity Allocation, ECCV 2020
> >
> > [R18] Discrete Model Compression with Resource Constraint for Deep Neural Networks, CVPR 2020

---

> > > ### Author Response · Authors · 2022-08-08
> > > **We would like to clarify the following:**
> > >
> > > 1. Mobilenet V1 does not contain residual connections.  That is one of the core differences between V1 and V2.  Please see reference [38] in the manuscript.
> > >
> > > 2. According to a recent pruning survey [6]: “the most common combination of dataset and architecture [is] VGG-16 on ImageNet”.
> > >
> > > 3. According to Table 4 in the Appendix of [R4], the number cited as SOTA is reported after 300 epochs of global fine tuning.  We report results before global fine tuning.  Therefore, we would expect them to be different. In addition, Section 3.1 of [R4] notes that tensor-decomposition based methods struggle on Mobilenet due to its architecture.
> > >
> > > 4. In [R8], the numbers you cite (Table 2, last method in VGG-16 section) are reported after the same amount of fine-tuning as was used to train the original model.  Pre-fine-tuning results are not reported.  Figure 2 in our manuscript only includes pre-global-fine-tuning results.
> > >
> > > 5. We employ the compression step of LRank as implemented by reference [51], but do not employ the modified training in our experiments.
> > >
> > > 6. As highlighted by [50], it is important to evaluate compression/pruning strategies on metrics beyond accuracy.

---

> > > > ### Comment · Reviewer_12sE · 2022-08-09
> > > > **Clarification is limited**
> > > >
> > > > [6] is a dated paper that summarizes the experiments before 2020. Please refer to papers published at NeurIPS. Only results for VGG-16 are totally NOT fair to other papers [R5-R18]. Even for the VGG, your results are still hugely inferior to SOTAs. Why not run it with the same settings of SOTAs, showing your method practically works well?
> > > >
> > > > Even with solid theoretical content, I think this paper has obvious drawbacks, i.e., practical performance. Letting ACs make the final decision is a good choice.

---

### Official Review · Reviewer_eWGv · 2022-07-10

**Rating:** 8
**Confidence:** 4
**Soundness:** 4 excellent
**Presentation:** 3 good
**Contribution:** 4 excellent

**Summary:**

This paper proposes a pruning method that uses interpolative decomposition of matrices to prune neurons/channels of a model. Generalization bound for the pruned model is provided for the two-layer network case. The method shows good pre-finetuning, post-finetuning quality metrics and good correlations with the un-pruned models.

**Questions:**

To successfully approximate the pre-compression layer, it seems that a good pruning data set is crucial. Would it be possible to show some empirical results on the dependence on the number of pruning sample?

**Strengths And Weaknesses:**

Strengths:

1. The paper proposes a novel approach to use interpolative decomposition to compress learned linear/convolutional layers and propagate compression corrections to other layers. The idea of using interpolative decomposition is new and solid;

2. The authors provided theoretical analysis on generalization bounds of a compressed two-layer network;

3. The experiments are done carefully and multiple metrics are used to compare relative performance of different compression methods;

4. The authors provided theoretical results on dependence on the number of samples for the two-layer network case.

Weaknesses:

For deeper networks, the paper didn't offer discussion on dependence on the number of samples.

---

> ### Author Response · Authors · 2022-08-02
> **Response to reviewer eWGv**
>
> Thank you for the helpful feedback. This is an insightful question: we provide results from two additional experiments to illuminate our method’s sensitivity (or lack thereof) to the prune set.
>
> First, we compare the pre-fine-tuning accuracy as a function of the pruning set size. Please consult our updated Appendix I for the figures. For ImageNet the benefit of increasing the prune set quickly tapers off. In fact, we see no meaningful change in pre-fine-tuning accuracy between using a prune set of 5k to 10k. On CIFAR-10 we are able to perform additional experiments that show this trend is consistent across the degree of compression targeted. We see no meaningful difference in the pre-fine-tuning accuracy for a prune set of size 0.5k, 1k, and 1.5k, across a range of compression targets.
>
> It is important to note that the number of examples in the pruning set must be larger than the number of neurons (or channels) that we expect to prune to for each layer. For example, with a VGG-16 network on CIFAR-10 we need at least 500 images in order to prune the last layer. Earlier layers can be pruned with fewer images, because distinct patches from the same image can act as multiple “examples”.
>
> Second, we show that our method is robust to a non-representative pruning set. We exclude a class from the prune set, and show that our compressed model still performs well across all classes (including the one not used for pruning). Please consult our updated Appendix I for the figure. We have a VGG16 model $M$ trained on CIFAR10. We remove a class from the pruning set, but leave it in the train and test sets. We then prune the model $M$ using our ID-based method.  In Figure 14 of Appendix I we show the per-class accuracies as a function of amount pruned. Note we don’t perform any fine-tuning. As a comparison we prune the model $M$ with magnitude pruning, also recording in Figure 15 of Appendix I the per-class accuracies. We then fine tune, using only images from the 9 classes.  While the magnitude pruned model recovers on the represented classes, it fails to recover any accuracy for the unrepresented class.
>
> We see that our ID-based method is able to prune with under-representative data and still preserve the per-class accuracies, including the class removed from the pruning set. Conversely, magnitude pruning cannot recover the accuracy of the missing class. We conjecture that pruning methods such as ours with high correlation to the original model–which therefore preserve the underlying decision boundaries in regions around valid data points–are more likely to preserve the accuracy of the unrepresented class.

---

### Official Review · Reviewer_s68B · 2022-07-10

**Rating:** 5
**Confidence:** 2
**Soundness:** 3 good
**Presentation:** 3 good
**Contribution:** 2 fair

**Summary:**

The main contribution of the paper is a new model compression method that simultaneously 1) preserves the original model's per-example decisions, 2) maintains the network's structure, 3) determines the per-layer compression levels automatically, and 4) eliminates the need for retraining the compressed model. The authors demonstrate the efficacy of their method through extensive experiments on CIFAR-10 and ImageNet datasets.

**Questions:**

I listed some questions in the previous section. My biggest concern is that authors use "preserving the model correlation" as one of the main criteria in model compression by claiming that this implies improved adversarial robustness, fairness, and sub-class accuracy. This relation is never made clear in the paper and I don't see why this is true.

**Limitations:**

The authors addressed the limitations.

**Strengths And Weaknesses:**

I generally like the approach, and the goal of the paper, and find the analyses theoretically sound. My only hesitation is their claim that preserving the model's per-example decisions would boost fairness and adversarial robustness. If we look at the "average" correlation between the compressed and original model, we will still miss errors in underrepresented data samples. So I am not sure how preserving the correlation brings an advantage over preserving the accuracy of the model in terms of fairness. The authors also mention adversarial robustness. Again, I could not make the connection between preserving the model correlation and adversarial robustness. Can the authors explain the relation and make it more clear in the paper?

I find the paragraph between lines 47-57 a bit confusing. What does "an independently trained VGG16 model" refer to?

---

> ### Author Response · Authors · 2022-08-02
> **Response to reviewer s68B**
>
> Thank you for the helpful feedback. We will respond to the specific questions below. Primarily, we will explain how model correlation serves as a generic metric for model preservation.
>
> Q: *Does preserving per-example decisions boost fairness or adversarial robustness?* \
> A:  We would like to clarify that we do not believe preserving per-example decisions “boosts” any of these properties. Rather, we strive to match per-example decisions in an effort to preserve properties of the baseline model—if a model has been trained to be fair or robust (in whatever sense is appropriate) we would like to be able to prune a model while maintaining these qualities to the extent possible.
>
> Q: *Why does preserving correlation give an advantage over preserving accuracy? How does the correlation metric connect to preserving fairness or robustness?* \
> A: Thanks for asking this question; it is a key point which we want to communicate with our paper.
>
> First, we will explain the benefit of our correlation metric over accuracy in measuring model preservation. Take a pretrained model $M$, a resulting compressed model $M_C$, and an evaluation set $(X,Y)$. Accuracy measures the similarity between $Y$ and $M_C(X)$. It only tells us how many examples $M_C$ correctly classified; it gives no connection between the learned functions of $M$ and $M_C$. Instead our correlation metric measures the similarity between $M(X)$ and $M_C(X)$. Correlation is an aggregate measure of how closely $M_C$ preserves the per-example decisions of $M$. In addition, correlation captures both what $M$ classified correctly and incorrectly. Accuracy only captures what is correctly classified.
>
> Second, we explain the connection between our correlation metric and fairness/robustness. Our main claim here is that by better preserving a model’s pre-example decisions, we can better preserve special properties of the model. Fairness and robustness are two specific examples of such special properties. However we propose model correlation as a generic metric to measure preservation between a pre-trained and compressed model. There are specific metrics one could employ, but that becomes a specialized task of preserving a specific model property.
>
> We further conduct an experiment to illustrate the connection between model correlation and a fairness metric of per-class accuracy. Please consult our updated Appendix I for the figure. We have a VGG16 model $M$ trained on CIFAR10. We remove a class from the pruning set to simulate an under-represented class (but leave it in the train and test sets). We then prune the model $M$ using our ID-based method, seen in Figure 14 of Appendix I. We plot the model correlation and per-class accuracies as a function of amount pruned. Note that we don’t perform any fine-tuning. As a comparison we prune the model $M$ with magnitude pruning, as seen in Figure 15 of Appendix I.  We also plot the model correlation and per-class accuracies.  While the magnitude pruned model recovers on the represented classes, it fails to recover any accuracy for the unrepresented class.
>
> We see that our ID-based method is able to prune with under-representative data and still preserve the correlation with the original model $M$ trained on representative data. Our ID-based method is also able to preserve the per-class accuracy: our measure of fairness in this example. Conversely, magnitude pruning performs worse in correlation and also cannot recover the accuracy of the missing class even using fine-tuning unless it is given examples from the underrepresented class during fine-tuning. We conjecture that pruning methods with high correlation to the original model–which therefore preserve the underlying decision boundaries in regions around valid data points–are more likely to preserve the accuracy of the unrepresented class.
>
> Q: *Lines 47-57: what does an independently trained model mean?* \
> A: We are happy to update the language in the main text to make this more clear.
> Consider a pre-trained model A. Another model B of the same architecture is “independently trained” if during training no information from model A’s parameters is passed to model B’s parameters. Essentially this means retraining the same architecture from scratch twice to obtain models A and B. The correlation between these two independently trained models captures the natural variation in the learned function over the data.

---

> > ### Comment · Reviewer_s68B · 2022-08-08
> > **Thanks for the detailed response.**
> >
> > I thank the authors for the detailed explanation of the "correlation" metric. After reading the rebuttal, I am now more convinced about the authors' claim for preserving fairness through this metric.

---

### Author Response · Authors · 2022-08-02
**General response to all reviewers:**

We thank the reviewers for taking the time to provide us with helpful feedback. Please find specific responses to individual questions/concerns below in a comment on each review.

From the comments below we’ve decided to develop and share these additional experimental results which can be found in our updated Appendix I:

1. The sensitivity to the prune set size

2. The robustness to removing a class from the prune set

---

### Meta-Review · Area_Chair_5w3C · 2022-08-27

**Recommendation:** Accept
**Confidence:** Less certain

**Metareview:**

The submission proposes an interpolative decomposition scheme for neural network compression to reduce FLOPs and number of parameters at a reduced cost in accuracy/faithfulness to the original model. The authors provide theoretical evidence, albeit in the two layer case, and empirical evidence on a set of architectures and datasets of the soundness of their claims. While the most negative review (12sE) contained several inaccuracies that misrepresented the submission, discussions with reviewers have nonetheless helped clarified several points in their paper to the point that the majority of reviewers were satisfied with the submission.
Therefore, I recommend this paper for acceptance.

**Award:**

No

---

### Decision · Program_Chairs · 2022-09-14

Accept